# Novel Fused Core Chromophore Incorporating Spirofluorene and Anthracene Groups for Sky-Blue Emission and Solution-Processed White Devices

**Sangwook Park †, Seokwoo Kang †, Sunwoo Park, Hyukmin Kwon, Hayoon Lee, Kiho Lee and Jongwook Park ***

Integrated Engineering, Department of Chemical Engineering, Kyung Hee University,
Yongin-si 17104, Gyeonggi, Republic of Korea; pswook@khu.ac.kr (S.P.); swkang@khu.ac.kr (S.K.);
mdrafix@khu.ac.kr (S.P.); hm531@khu.ac.kr (H.K.); kssarang1@khu.ac.kr (H.L.); kiholee@khu.ac.kr (K.L.)
* Correspondence: jongpark@khu.ac.kr; Tel.: +82-10-8759-8485
† These authors contributed equally to this work.

**Abstract:** New blue-light-emitting materials, 2,7-Bis-[1,1′;3′,1″]terphenyl-spiro-fluorene[3,4]naphthalene (TP-AFF-TP) and spiro-fluorene[3,4]-5-terphenylnaphthalene (TP-ASF) were synthesized based on a fused core with anthracene and spirofluorene. The photoluminescence (PL) maximum values of TP-AFF-TP and TP-ASF in film states exhibited 477 nm and 467 nm within the blue region, respectively. Degradation temperature ($T_d$) values for TP-AFF-TP and TP-ASF were very high at 481 °C and 407 °C, respectively. TP-AFF-TP and TP-ASF exhibited power efficiencies (PE) of 1.03 lm/W and 2.39 lm/W, respectively, along with luminance efficiencies (LE) of 2.55 cd/A and 5.17 cd/A, respectively, in nondoped organic light-emitting diode (OLED) devices in which the newly synthesized compounds were employed as emissive layers. The achieved CIE values were (0.25, 0.45) for TP-AFF-TP and (0.17, 0.31) for TP-ASF. Furthermore, when TP-ASF was utilized as one of the emissive materials in solution-processed white OLED devices, the resultant device showcased a notably high LE of 3.13 cd/A, a PE of 2.69 lm/W, and a white CIE value of (0.30, 0.34).

**Keywords:** blue-light-emitting materials; fused core moiety; anthracene; spirofluorene groups

## 1. Introduction

Ever since the discovery of electroluminescence (EL) performance in organic materials by Tang et al. in 1987, extensive research has been conducted on organic semiconductors. This research covers various application areas, such as organic light-emitting diodes (OLEDs) [1–3], organic thin-film transistors (OTFTs) [4], and organic photovoltaics (OPVs) [5]. As of now, researchers have developed and explored numerous conjugated organic molecules to showcase EL across the spectrum, covering red, green, and blue wavelengths. In the pursuit of crafting full-color OLED displays, the demand for high-performance emitters in red, green, and blue spectrums has been paramount. These emitters should possess exceptional EL efficiency, robust thermal characteristics, extended device lifetime, and precise color purity as defined by the 1931 Commission Internationale de l'Eclairage x, y coordinates (CIE (x, y)). Creating blue light emitters focuses on addressing the challenges of high efficiency and extended device lifetime, which are typically hindered by their inherently wide band gaps. The significant band gap of blue-emitting materials engenders a mismatched recombination zone for electrons and holes, resulting in diminished EL efficiency and a shorter operational lifespan for OLED devices. Over time, an array of blue-light emitters has been extensively explored, including derivatives based on anthracene [6–10], fluorine [11], pyrene [12], and di(styryl)arylene [13] with the aim of enhancing their electroluminescent properties. These chromophores have been extensively studied as emitting materials, and the structures that could have been developed using a single chromophore have reached their limits. Subsequently, advancements have led to the

development of luminophores composed of dual-core and triple-core structures, and further progress has been made by employing fused core combinations [14,15]. Recently, there have also been many reports about multiple resonance emitters and hyperfluorescence blue OLEDs [16,17].

Traditionally, vapor deposition has been the mainstream method for OLED device commercialization due to its guaranteed high performance. However, the solution process of OLED devices has been greatly focused because of the simple process and particularly low-cost effect, although solution processing still needs to develop the related device property [18–21]. While the utilization of polymer-based light-emitting diodes (PLEDs) is a feasible avenue for the solution processing of OLEDs, challenges arise in terms of effectively purifying PLED materials and regulating molecular weight during large-scale production. On the contrary, solutions based on small molecules for the procedure need to demonstrate harmony between remarkable dissolvability and increased radiation effectiveness attributes.

Based on this information, we designed new sky-blue-emitting materials, including anthracene and spirofluorene core, for suitable solution processes. First, we prepared the main core with a fused structure of anthracene and spirofluorene for good luminance efficiency (LE). Second, to prevent π-π stacking interactions, a side group was introduced into the core structure. Also, in the case of the two fused chromophores, it is confirmed that the EL performance is improved upon the side group position of anthracene or spirofluorene. Lastly, highly twisted compounds provide good solubility and they can be applicable to the solution process. The newly synthesized blue-emitting materials consist of 2,7-Bis-[1,1′;3′,1″]terphenyl-spiro-fluorene[3,4]naphthalene (TP-AFF-TP) and fluorene[3,4]-5-terphenylnaphthalene (TP-ASF) (Figure 1). The synthesized materials were analyzed thermal and electronic characteristics through techniques such as thermogravimetric analysis (TGA), ultraviolet-visible light absorption spectroscopy (UV-Vis), and photoluminescence (PL) spectroscopy. Using these synthesized materials, multilayered electroluminescent (EL) devices were manufactured. These materials were utilized not only as emitters without doping but also as dopants in white OLEDs based on solution processing [22].

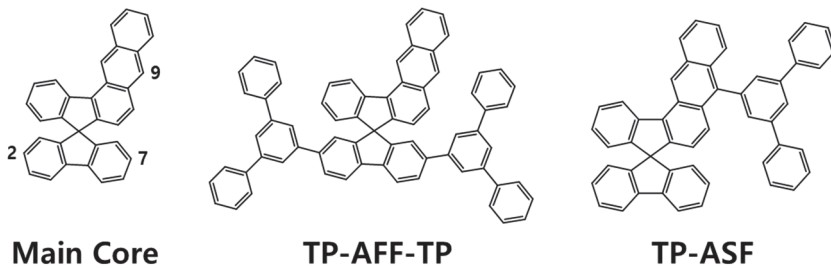

**Figure 1.** Chemical structures of fused new core and the synthesized materials.

## 2. Materials and Methods

### 2.1. General Information

The synthetic products in this article were confirmed by Bruker Avance 300 for proton NMR spectroscopy(Bruker, Billerica, MA, USA), and JEOL JMS-AX505WA (JEOL Ltd., Tokyo, Japan), HP5890 series II (Hewlett Packard, Wokingham, UK)for fast atom bombardment (FAB) mass spectroscopy. The optical properties were evaluated using HP 8453 UV-VISNIR spectrometer (Agilent, Santa Clara, California, USA) for UV-Vis absorption, and Perkin-Elmer luminescence spectrometer LS50 (Perkin-Elmer, Waltham, MA, USA) (Xenon flash tube) for PL and EL. TGA offered degradation temperatures ($T_d$). Non-doped OLED devices for blue emission were built in the structure of indium tin oxide (ITO)/2-TNATA (60 nm)/NPB (15 nm)/synthesized material (35 nm)/Alq3 (20 nm)/LiF (1 nm)/Al (200 nm): 2-TNATA stands for 4,4′,4″-tris(N-(2-naphthyl)-N-phenylamino)-triphenylamine, and constituted the hole injection layer (HIL); N,N′-bis(naphthalen-1-yl)-N,N′-bis(phenyl)benzidine (NPB) made up the hole transporting layer (HTL); Tris(8-hydroxyquinolinato)aluminium ($Alq_3$) formed the electron transporting layer (ETL); the

electron injection layer (EIL) comprised lithium fluoride (LiF); and ITO served as the anode, while aluminum (Al) was employed as the cathode. Organic molecules were thermally evaporated at a vacuum base pressure of $10^{-6}$ Torr, and formed an emitting layer of 4 mm$^2$ with a deposition rate of 0.1 nm/s. The Al layer was also formed in pressure of $10^{-6}$ Torr. A white OLED device by solution processing in the structure of ITO/PEDOT:PSS (40 nm)/TP-ASF: 0.2 or 0.4 wt% rubrene (50 nm)/TPBi (30 nm)/LiF/Al. PEDOT:PSS was applied as HTL, and TP-ASF was used as a blue dopant as well as a host for rubrene. The role of a red dopant was played by (5,6,11,12)-Tetraphenylnaphthacene (rubrene). Chlorobenzene was chosen for a solvent. PEDOT/PSS mixtures (Baytron PVP CH8000, H. C. Starck GmbH) in a water dispersion were spin-coated over ITO anodes in the atmosphere. The spin-coated films were subjected to baking in ambient conditions on a hot plate at 110 °C for 5 min, followed by baking at 200 °C for 5 min in a nitrogen ($N_2$) environment. After spin-coating to achieve a thickness of 50 nm under a nitrogen atmosphere, the emitting layer was subjected to baking on a hot plate at 80 °C for 30 min. Electron-transporting layers made of TPBi, the deposition of LiF and Al layers was carried out under the same condition as applied to the nondoped OLED devices. The current-voltage (I-V) characteristics of the fabricated OLED devices were assessed using a Keithley 2400 Source Meter (Keithley, Cleveland, OH, USA). The EL spectrum of the devices was measured by a Minolta CS-1000A spectroradiometer (Konica Minolta, Toyko, Japan).

*2.2. Synthesis*

### 2.2.1. 5-Bromoanthraquinone (**1**)

Tert-butyl nitrite (12 mL, 100 mmol) was put into the suspension of 5-Aminonanthraquinone (15.0 g, 67.2 mmol) and Copper(II) bromide (18.0 g, 80.6 mmol) in $CH_3CN$ 300 mL while vigorously stirring, and then heating the mixture took place at 65 °C for a duration of 2.5 h, before the previously heated mixture was allowed to cool to room temperature (rt). Water (100 mL) was added and hydrochloric acid (3 M, 100 mL) was continuously added and stirred for an additional 20 min. After that, the mixture was filtered, cleaned with water (4 times each 100 mL) and EtOH (200 mL), and dried under a vacuum condition. Using chloroform as the eluting agent, the material was filtered through a silica gel plug, and finally was dried in vacuo yielding 13.7 g of 5-bromoanthraquinone as an orange powder: (88% Yield) $^1$H-NMR (300 MHz, CDCl$_3$) δ(ppm): 8.39–8.38 (d, 1H), 8.36 (d, 1H), 8.33 (d, 1H), 8.08–8.07 (d, 1H), 7.85–7.77 (m, 2H), 7.61–7.56 (t, 1H).

### 2.2.2. 1-Bromoanthracene (**2**)

We introduced 5-bromoanthraquione (6.0 g, 20.8 mmol) in acetic acid (200 mL) into a 3-neck round flask, and hydrogen bromide (30 mL) and hypophosphorous acid (40 mL) was sequentially added under agitation. After allowing the reaction to continue for 4 days at 120 °C, distillation was conducted in order to remove the solvent, and the remainder was filtered using chloroform as an eluent and dried in vacuum obtaining 2.3 g of 1-bromoanthracene as a yellow powder: (43% Yield) $^1$H-NMR (300 MHz, CDCl$_3$) δ(ppm): 8.81 (s, 1H), 8.43 (s, 1H), 8.11–7.96 (m, 3H), 7.80–7.77 (d, 1H), 7.55–7.44 (m, 2H), 7.31–7.25 (t, 1H).

### 2.2.3. 1-Anthracene Boronic Acid (**3**)

We dissolved 1-Bromoanthracene (5.18 g, 20.1 mmol) in anhydrous THF (150 mL), and n-BuLi (12 mL, 24.1 mmol, 2.0 M in cyclohexane) was then slowly added at −78 °C. After an hour, triethyl borate (4.8 mL, 28.2 mmol) was added to the mixture, and it was slowly heated up to rt and kept being stirred for 12 h. After adding distilled water, the product was extracted with ethyl acetate. Magnesium sulfate was added for drying and was followed by filtration and distillation under reduced pressure, and column separation yielded finally 1-Anthracene boronic acid: (60% Yield) $^1$H-NMR (300 MHz, DMSO) δ(ppm): 8.98 (s, 1H), 8.56 (s, 1H), 8.48 (s, 2H), 8.11–7.96 (m, 3H), 7.80–7.77 (d, 1H), 7.55–7.44 (m, 2H), 7.31–7.25 (t, 1H).

### 2.2.4. 1-(2-Bromo-phenyl)-anthracene (**4**)

We introduced 1-Anthracene boronic acid (2.9 g, 13.0 mmol), 1-bromo-2-iodobenzene (2.35 mL, 18.3 mmol), tetrakis(triphenylphosphine)palladium(0) (0.42 g, 0.36 mmol) into a 3-neck round flask, and mixed with anhydrous THF (80 mL). It was heated under reflux for a day after 2 M potassium carbonate 8 mL was added to the mixture at 60 °C. The product was extracted by using chloroform and water. The remaining water in the organic layer was eliminated by $MgSO_4$. Column refinement under MC: hexane = 1:10 eluent was proceeded after evaporation of product in the organic layer. The product obtained through column filtration was dissolved in chloroform and then reprecipitated using methanol prior to the filtration step. By evaporating the filtered material, 2.9 g of white solid were obtained: (58% Yield) [1]H-NMR (300 MHz, $CDCl_3$) δ(ppm): 8.50 (s, 1H), 8.09–8.06 (d, 1H), 8.02–7.99 (d, 2H), 7.86–7.83(d, 1H), 7.81–7.78 (d, 1H), 7.55–7.36 (m, 7H).

### 2.2.5. 2,7-Dibromo-spiro-fluorene[3,4]naphthalene (**5**)

We dissolved 1-(2-Bromo-phenyl)-anthracene (1.26 g, 3.78 mmol) in anhydrous THF (10 mL), and n-BuLi (2 mL, 4.16 mmol, 2.0 M in cyclohexane) was slowly added to it at −78 °C. One hour later, 2,7-Dibromo-fluorenone (1.66 g, 4.91 mmol) was added. After gradually heating up to rt, it was stirred for an additional 3 h. It is extracted with MC after adding sodium bicarbonate. After evaporation of the remaining solvent, HCl (1.1 mL, 12 M) and 11 mL of acetic acid were added to the raw product. The mixture was heated under reflux overnight. After the reaction was completed, the product was extracted by using chloroform and water. The organic layer containing a small amount of residual water was treated with $MgSO_4$ to remove it. Through a silica gel column (EA/*n*-hexane, 1:10), the crude product was purified to obtain (5) compounds as yellow powder. [1]H-NMR (300 MHz, $CDCl_3$) δ(ppm): 9.38 (s, 1H), 8.65–8.63 (d, 1H), 8.51 (s, 1H), 8.23–8.20 (d, 1H), 8.06–8.03 (d, 1H), 7.85–7.82 (d, 1H), 7.73–7.70 (d, 2H), 7.61–7.50 (m, 5H), 7.22–7.19 (d, 1H), 6.86–6.83 (d, 3H), 6.77–6.74 (d, 1H).

### 2.2.6. Spiro-fluorene[3,4]naphthalene (**6**)

We dissolved 1-(2-Bromo-phenyl)-anthracene (1.43 g, 4.29 mmol) in anhydrous THF (10 mL) and n-BuLi (2.57 mL, 5.14 mmol, 2.0 M in cyclohexane). After fluorenone (1.43 g, 5.57 mmol) was added, the mixture was slowly heated up to rt and kept being stirred for 3 h. After adding sodium bicarbonate, it was extracted with MC. After evaporation of the remaining solvent, HCl (1.2 mL, 12 M) and 12 mL of acetic acid were added to the raw product. The mixture was heated under reflux overnight. After the reaction was completed, the product was extracted by using chloroform and water. The organic layer containing a small amount of residual water was treated with $MgSO_4$ to remove it. Column refinement under EA: hexane = 1:10 eluent was proceeded after evaporation of product in the organic layer in order to obtain (6) compounds as yellow powder. [1]H-NMR (300 MHz, $CDCl_3$) δ(ppm): 8.44 (s, 1H), 8.26–8.24 (d, 1H), 8.19–8.17 (d, 1H), 8.03–7.97 (m, 3H), 7.90–7.87 (d, 1H), 7.62–7.57 (t, 1H), 7.41–7.35 (t, 2H), 7.22–7.17 (q, 3H), 7.13–7.03 (t, 2H), 6.92–6.83 (m, 4H), 6.49–6.46 (d, 1H).

### 2.2.7. Spiro-fluorene[3,4]-5-bromonapthalene (**7**)

Spiro-fluorene[3,4]naphthalene (0.5 g, 1.20 mmol) was dissolved in chloroform 10 mL, and bromine (0.06 mL, 1.11 mmol) was slowly added at 0 °C. The solution was stirred for 3 h. By adding acetone, the mixture was quenched, and then extracted using chloroform and water. Using $MgSO_4$ for removing a small amount of water in the organic layer. Reprecipitation with methanol and chloroform was used to obtain a yellow powder: (77% Yield) [1]H-NMR (300 MHz, $CDCl_3$) δ (ppm): 9.45 (s, 1H), 8.63–8.61 (d, 1H), 8.57–8.53 (d, 1H), 8.41–8.37 (d, 1H), 8.22–8.19 (d, 1H), 7.91–7.89 (d, 2H), 7.66–7.52 (m, 3H), 7.43–7.37 (t, 2H), 7.25–7.17 (t, 1H), 7.13–7.08 (t, 2H), 6.91–6.85 (t, 2H), 6.74–6.72 (d, 2H).

2.2.8. 2,7-Bis-[1,1′;3′,1″]terphenyl-spiro-fluorene[3,4]naphthalene (TP-AFF-TP)

We put 2,7-Dibromo-spiro-fluorene[3,4]naphthalene (0.26 g, 0.45 mmol), 4,4,5,5-tetramethyl-2-[1,1′;3′,1″]terphenyl-5′-yl-[1,3,2]dioxaborolane (0.35 g, 0.99 mmol), tetrakis(triphenylphosphine)palladium(0) (0.05 g, 0.045 mmol) in a 3-neck round flask, and anhydrous toluene (40 mL) and 2 M sodium hydroxide 20 mL were added. The mixture was heated under reflux overnight. After the reaction was completed, the solution was subjected to extraction using chloroform and water. The product was extracted by using chloroform and water. The organic layer containing residual water was treated with $MgSO_4$ to remove it. Column refinement under EA: hexane = 1:4 eluent was proceeded after evaporation of product in the organic layer. Chloroform and methanol were used for reprecipitation and then, by evaporating the filtered solid, 0.12 g of white power was obtained: (30% Yield) [1]H-NMR (300 MHz, $CDCl_3$) δ(ppm): 9.56 (s, 1H), 8.89–8.86 (d, 1H), 8.65 (s, 1H), 8.43–8.40 (d, 1H), 8.32–8.30 (d, 2H) 8.10–8.08 (d, 1H), 7.99–7.92 (t, 3H), 7.70–7.66 (t, 10H), 7.61–7.52 (m, 7H), 7.42–7.37 (t, 8H), 7.34–7.29 (t, 4H), 7.28–7.25 (d, 1H), 7.1 (s, 2H), 6.92–6.85 (q, 2H). [13]C NMR (100 MHz, $CDCl_3$) δ(ppm): 142.35, 139.00, 138.83, 138.44, 138.27, 135.43, 130.73, 130.26, 129.54, 128.90, 128.76, 128.58, 128.19, 128.07, 127.30, 127.20, 126.73, 126.63, 125.83, 125.40, 124.87, 124.74, 124.09, 64.89. Fab[+]-MS: 873 *m/z*.

2.2.9. Spiro-fluorene[3,4]-5-terphenylnaphthalene (TP-ASF)

Spiro-fluorene[3,4]-5-bromonapthalene (0.27 g, 0.54 mmol), 4,4,5,5-tetramethyl-2-[1,1′; 3′,1″]terphenyl-5′-yl-[1,3,2]-dioxaborolane (0.23 g, 0.65 mmol), tetrakis(triphenylphosphine) palladium(0) (0.03 g, 0.02 mmol) were placed in a 3-neck round flask, and anhydrous toluene (25 mL) was added. 2 M sodium hydroxide 10 mL was added. The mixture was heated under reflux overnight. After reaction, the solution was extracted using chloroform and water. Using $MgSO_4$, the remaining water in the organic layer was removed. Column refinement under chloroform: hexane = 1:3 eluent was proceeded after evaporation of product in the organic layer. Chloroform and methanol were used for re-precipitation and then, by evaporating the filtered solid, 0.17 g of white power was obtained: (48% Yield) [1]H-NMR (300 MHz, THF) δ(ppm): 9.65 (s, 1H), 8.84–8.81 (d, 1H), 8.39–8.37 (d, 1H), 8.07–8.06 (t, 1H), 7.92–7.90 (d, 2H), 7.78–7.76 (q, 5H), 7.74–7.69 (d, 2H), 7.62–7.53 (q, 3H), 7.44–7.27 (m, 9H), 7.19–7.15 (t, 1H), 7.08–7.05 (t, 2H), 6.82–6.78 (d, 1H), 6.71–6.66 (d, 2H), 6.64–6.60 (d, 1H). [13]C NMR (100 MHz, THF-d[8]) δ(ppm): 142.04, 141.63, 140.01, 139.24, 137.48, 136.37, 136.31, 135.07, 133.73, 133.56, 131.67, 130.18, 129.33, 128.08, 127.85, 127.84, 127.71, 127.63, 127.15, 126.95, 126.78, 126.52, 126.14, 125.56, 125.03, 124.29, 123.80, 64.28. Fab+-MS: 644 *m/z*.

## 3. Results and Discussion

### 3.1. Molecular Design, Synthesis, and Characterization

A new core chromophore was prepared based on anthracene and spirofluorene, both of which exhibit remarkable luminance efficiency. This chromophore has a new core structure based on anthracene and spirofluorene moieties. Ring-fused structures have attracted much attention because of their rigid and nonextended π-conjugated properties. This ring-fused structure has advantages such as intense luminescence and high thermal stability [23,24]. Based on this approach, we compared the EL efficiency property when side groups were changed with the different position of the new core as well as different substitution number of side groups. Since anthracene core moiety has high electron density, the position nine of anthracene in the core group was selected for the link position of the side group as well as the positions two and seven of spirofluorene in the core group (see Figure 1). Especially, the position nine of anthracene can provide the increased EL efficiency due to the dense electron distribution of anthracene. Regarding the molecular design, the incorporation of bulky terphenyl (TP) side groups into the fused core aims to prevent π-π stacking interactions. This modification can enhance solubility and disrupts the formation of excimers. The chemical structures are presented in Figure 1, while the corresponding pathways of synthesis are visually represented in Scheme 1. Column purification and

reprecipitation were used to purify these synthesized compounds. Also, these materials were analyzed by NMR, FAB-MS analysis to obtain pure compounds.

**Scheme 1.** Synthetic pathways of synthesized materials.

### 3.2. Photophysical Properties

Table 1 and Figure 2 provide a comprehensive summary of the UV-Vis absorption and photoluminescence (PL) spectra of the synthesized compounds in both solution and film states. In the solution state, TP-AFF-TP and TP-ASF exhibit UV maximum values of 410 and 419 nm and PL maximum values of 426, 447 nm and 439, 456 nm, respectively. The solution FWHM values were at 55 and 57 nm, respectively. TP-ASF has UV and PL maximum value in the red-shifted region more than that of TP-AFF-TP. Also, TP-ASF had a slightly broader FWHM compared to TP-AFF-TP. Through density functional theory (DFT), it is confirmed that most electron density exists on anthracene in HOMO and LUMO energy levels for TP-AFF-TP and TP-ASF. In the case of TP-ASF, the substitution of the side group on the anthracene moiety, which possesses a higher electron density, might have led to lengthen the intramolecular conjugation and the generation of numerous electron transition states, potentially contributing to a large Stoke-shift result (see Figure 3). In the film state, the UV maximum values of TP-AFF-TP and TP-ASF show 416 nm and 425 nm, respectively. There was a difference of 6 nm for both the solution and film state. Moreover, the photoluminescence (PL) spectra of TP-AFF-TP and TP-ASF, when in a film state, displayed red shifts of 30 nm and 11 nm, respectively, in contrast to their respective spectra in the solution state. The commonly observed phenomenon involves the red shifts in UV and PL peak wavelengths, along with broader full width at half maximum (FWHM) values, during the transition from a solution to a film state. This is attributed to the intermolecular interactions that arise from the reduced intermolecular distances in the film state. However, in contrast to the solution, TP-AFF-TP exhibited luminescence in the longer wavelength region compared to TP-ASF. In addition, FWHM values of TP-AFF-TP and TP-ASF exhibit 96 nm and 68 nm in the film state, respectively. This means that TP-AFF-TP has stronger intermolecular interaction than that of TP-ASF. This is because TP-ASF has its side group, TP, substituted on the anthracene moiety, which can suppress the planar

packing of anthracene. However, TP-AFF-TP has its side group already substituted on the highly twisted fluorene portion, and this might have been ineffective in preventing effective intermolecular π-π stacking. Therefore, incorporating a large side group at a strategically selected location on the central molecule can efficiently impede π–π stacking interactions. This approach effectively mitigates the possibility of emission suppression. Additionally, such alterations have the capacity to influence the quantum efficiency of photoluminescence (PLQE). As a result of PLQE measurement, PLQE values of TP-AFF-TP and TP-ASF were 45 and 88% in the solution state, and 20 and 53% in the film state. The values of the transition contribution and the oscillator strength through the time-dependent DFT (TD-DFT) calculation between HOMO and LUMO levels showed 83.8% and 0.117 in TP-AFF-TP and 91.7% and 0.262 in TP-ASF, respectively, as shown in Table S1. This causes TP-ASF with higher oscillator strength value to exhibit higher PLQE.

In order to measure the highest occupied molecular orbital (HOMO) values of the synthesized materials, ultraviolet photon spectroscopy (UPS) of AC-II and optical absorption spectroscopy were carried out. As indicated in Table 1, the highest occupied molecular orbital (HOMO) levels were measured at 5.80 eV and 5.72 eV, while the lowest unoccupied molecular orbital (LUMO) levels were recorded at 2.79 eV and 2.80 eV for TP-AFF-TP and TP-ASF, respectively, within the blue region. The band gaps of TP-AFF-TP and TP-ASF were 3.01 eV and 2.92 eV, respectively. Decomposition temperature ($T_d$) by TGA, which means thermal stability of the synthesized materials was measured. The $T_d$ values for TP-AFF-TP and TP-ASF were 481 and 407 °C, respectively. Glass transition temperature ($T_g$) of TP-AFF-TP and TP-ASF were measured by differential scanning calorimetry (DSC), which showed 204 and 190 °C, respectively, as shown in Figure S1. Under the joule heating, which comes from the device operation, this high thermal stability is able to stabilize the operation of the device.

**Table 1.** Photophysical properties of the synthesized materials.

| | Solution [a] | | | Film [b] | | | HOMO [d] (eV) | LUMO (eV) | Band Gap (eV) | $T_d$ (°C) |
|---|---|---|---|---|---|---|---|---|---|---|
| | $\lambda_{Abs}$ (nm) | $\lambda_{PL}$ (nm) | FWHM [c] (nm) | $\lambda_{Abs}$ (nm) | $\lambda_{PL}$ (nm) | FWHM [c] (nm) | | | | |
| TP-AFF-TP | 388,410 | 426, 447 | 55 | 393,416 | 477 | 96 | −5.80 | −2.79 | 3.01 | 481 |
| TP-ASF | 375, 396,419 | 439, 456 | 57 | 381, 401,425 | 467 | 68 | −5.72 | −2.80 | 2.92 | 407 |

[a] Chloroform solution ($1.00 \times 10^{-5}$ M). [b] Film with a thickness of 50 nm on glass. [c] FWHM of PL peak. [d] HOMO values derived from ultraviolet photoelectron spectra (Riken-Keiki, AC-2).

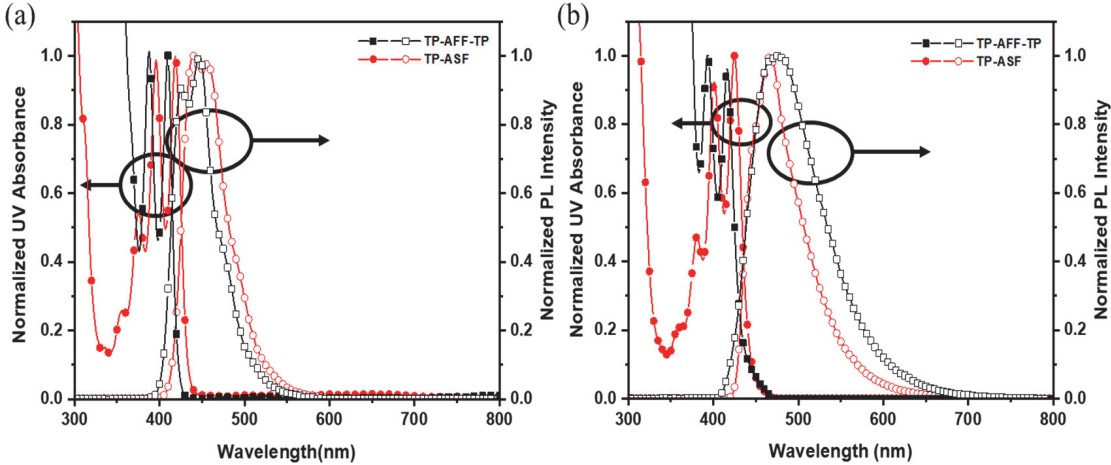

**Figure 2.** UV-visible absorption and photoluminescence analysis of synthesized substances (**a**) in the solution state (chloroform, $1 \times 10^{-5}$ M) and (**b**) in the vacuum-deposited film (thickness: 50 nm).

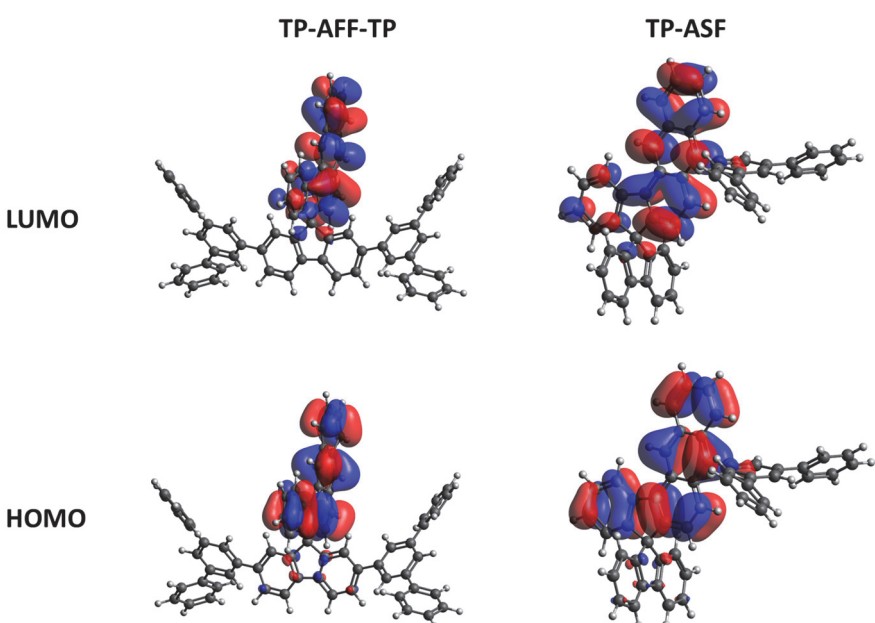

**Figure 3.** Electron density distribution of HOMO and LUMO in TP-AFF-TP and TP-ASF (calculated by the DFT/B3LYP/6-31G(d) for molecule optimization).

*3.3. Electroluminescence Properties*

The subsequent configuration of the nondoped device was used utilizing the synthesized materials: ITO/2-TNATA (60 nm)/NPB (15nm)/TP-AFF-TP or TP-ASF (35 nm)/Alq$_3$ (20 nm)/LiF (1 nm)/Al (200 nm). The EL performance and electrical properties are shown in Table 2 and Figure 4. The operating voltage of TP-ASF was approximately 0.9V lower than that of TP-AFF-TP. This can be attributed to the fact that in the band diagram, the hole injection from NPB and electron injection from Alq3 are more facile in TP-ASF than in TP-AFF-TP (see Figure 5a). The LE (cd/A) and power efficiency (PE) (lm/W) of the TP-ASF devices was high compared to TP-AFF-TP. TP-AFF-TP and TP-ASF exhibit luminance efficiencies of 2.55 cd/A and 5.17 cd/A, respectively, along with power efficiencies of 1.03 lm/W and 2.39 lm/W, respectively. IQE values of TP-AFF-TP and TP-ASF were 11 and 22%, respectively, in the solution state and 5 and 13%, respectively, in the film state. The external quantum efficiency (EQE) values of TP-AFF-TP and TP-ASF were 1.06 and 2.80%, respectively. All EL efficiencies of TP-ASF were higher than that of TP-AFF-TP. It might be that the side groups were substituted at the positions two and seven of spirofluorene, which has small electron density distribution, and at the position nine of anthracene, which has relatively large electron density distribution, as shown in Figure 3. As a result, TP-ASF, which has the side group on the electron-rich position, showed higher efficiency than TP-AFF-TP. These results show that the side group location, which is a connected core moiety, is important for an increase in the EL efficiency of the emitting material [25]. In addition, CIE values are (0.25, 0.45) and (0.17, 0.31) in cases of TP-AFF-TP and TP-ASF, respectively. This means TP-ASF had high color purity for blue by low CIEy value compared to TP-AFF-TP because introduced TP at anthracene effectively prevented π–π stacking. The reason why the EL maximum value is different from the PL maximum value is that the recombination zone under the device operation is not located at the center of the emitting layer inside. In order to solve this issue, we need to change the kind of hole-transporting layer (HTL) or electron-transporting layer (ETL) as well as the various thicknesses of HTL or ETL. Further studies on the engineering device regarding the different carrier transporting layers are underway. We will report it separately. Regarding the device lifetime (LT), the LT50 of TP-AFF-TP and TP-ASF were approximately 40 min and 1 h at 1000 nit, respectively. It is expected that the lifetime of the device would be increased if the charge balance were improved by applying different carrier transport materials based on the various thicknesses (Figure S2).

**Table 2.** EL performances of the fabricated OLED devices at 10 mA/cm².

| Materials | $V_{on}$ (V) | LE (cd/A) | PE (lm/W) | CIE (x, y) | $EL_{max}$ (nm) | FWHM (nm) |
|---|---|---|---|---|---|---|
| TP-AFF-TP | 8.41 | 2.55 | 1.03 | (0.25, 0.45) | 504 | 96 |
| TP-ASF | 7.50 | 5.17 | 2.39 | (0.17, 0.31) | 484 | 80 |

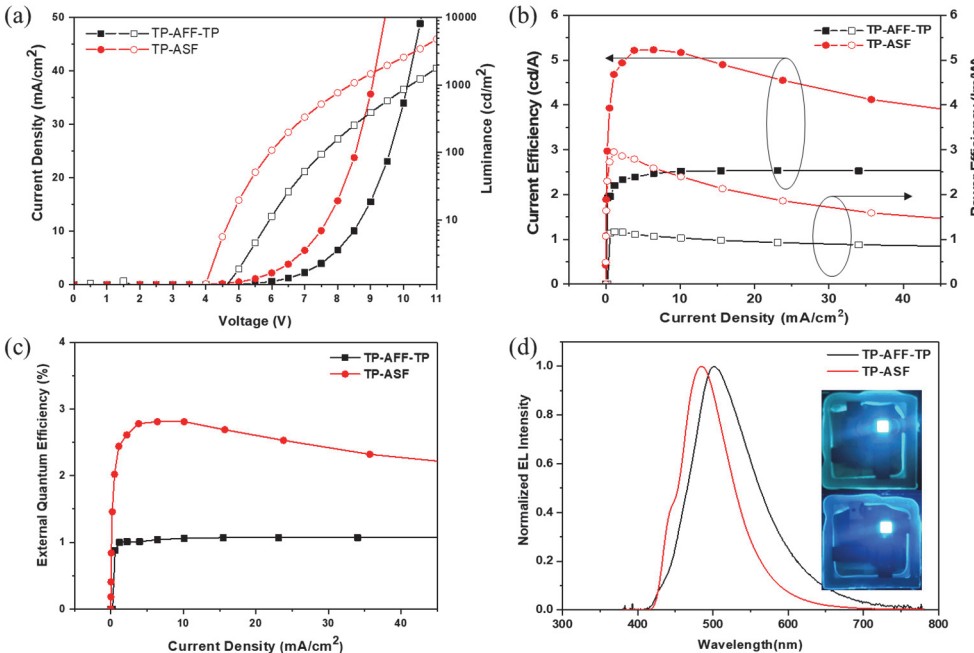

**Figure 4.** OLED device performance of TP-AFF-TP and TP-ASF emitters: (**a**) current-voltage-luminescence curve, (**b**) current efficiency and power efficiency according to the current density, (**c**) EQE and (**d**) EL spectra at 10 mA/cm²; and device photos of TP-AFF-TP (top) and TP-ASF (bottom).

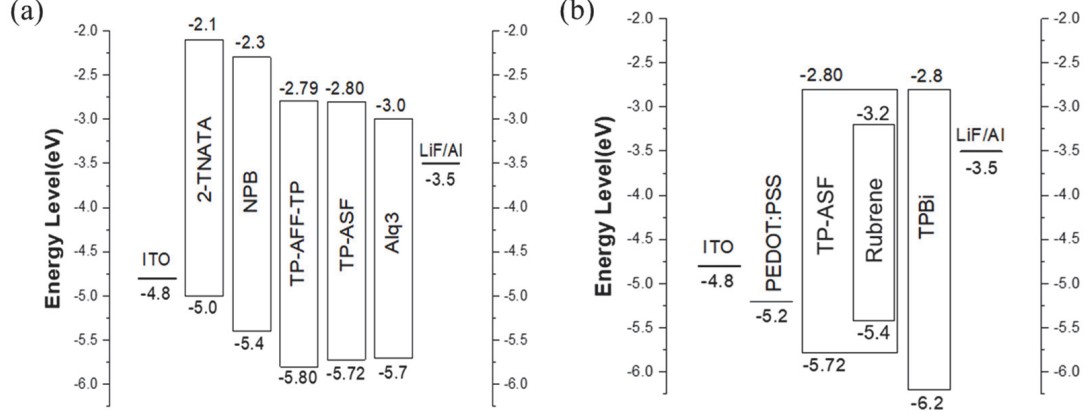

**Figure 5.** Device structure with the energy-level diagrams of (**a**) non-doped devices for TP-AFF-TP and TP-ASF, (**b**) white OLED using TP-ASF.

Recently, there has been a notable emphasis on solution-processed white OLEDs (WOLEDs) within the realms of OLED backlighting and lighting applications. The synthesized materials could be good candidates for solution processing. Based on the highly twisted molecular structure, it can increase the solubility in organic solvent. For example, a binaphthyl structure with a twisted backbone along an auxiliary axis exhibits good solubility, even in the case of polymers having a binaphthyl moiety [26]. Especially TP-ASF, which showed high efficiency in nondoped devices, was used as a blue dopant for WOLED. The

following configuration of WOLED device was fabricated using TP-ASF: ITO/PEDOT:PSS (40 nm)/TP-ASF: 0.2 wt% or 0.4 wt% rubrene (50 nm)/TPBi (30 nm)/LiF/Al. ETL, LiF, and Al electrodes were prepared under vacuum evaporation except PEDOT:PSS. The electroluminescent characteristics of the white OLED produced through solution processing are outlined in Table 3 and depicted in Figure 6. Operating voltages of the two devices were similar to 4.08 and 4.12 V because these two devices had the same structure except for the rubrene concentration (see Figure 5b). The white OLED devices of the 0.2 wt% rubrene (WOLED I) and 0.4 wt% rubrene (WOLED II) were showed LE of 3.13 cd/A, 4.24 cd/A and PE of 2.69 lm/W, 3.60 lm/W, respectively. Also, WOLED I and II showed EQE of 1.42 and 1.67%, respectively. The EL efficiency of WOLED II was improved more than that of WOLED I due to a high concentration of rubrene as the used dopant. The maximum values of EL spectra for the white OLED devices WOLED I and II were identical, measuring 463 nm and 549 nm, respectively. This means that the recombination zone was well generated within EML for both devices. However, WOLED I and II exhibited the CIE values of (0.30, 0.34), (0.36.0.40), respectively. These results mean that in terms of efficiency, WOLED II achieved high efficiencies, but in terms of color coordinates, WOLED I demonstrated a more suitable implementation for the ideal WOLED. In essence, WOLED I might be more applicable than WOLED II for commercial lighting display applications.

**Table 3.** EL performance of white OLED device using synthesized material: ITO/PEDOT:PSS (40 nm)/TP-ASF: 0.2 wt% or 0.4 wt% rubrene (50 nm)/TPBi (30 nm)/LiF (1 nm)/Al (200 nm) at 10 mA/cm$^2$.

| Materials | $V_{on}$ (V) | LE (cd/A) | PE (lm/W) | CIE (x, y) | $EL_{max}$ (nm) |
|---|---|---|---|---|---|
| TP-ASF: 0.2 wt% Rubrene | 4.08 | 3.13 | 2.69 | (0.30, 0.34) | 463, 549 |
| TP-ASF: 0.4 wt% Rubrene | 4.12 | 4.24 | 3.60 | (0.36,0.40) | 463, 549 |

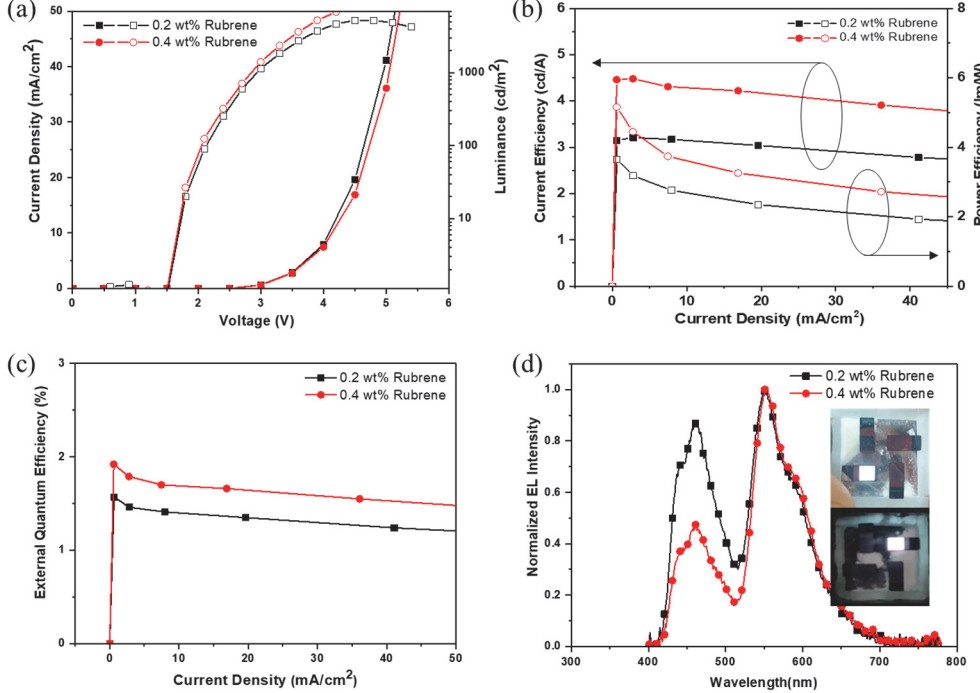

**Figure 6.** White OLED device performance with TP-AFF-TP and TP-ASF Emitters: (**a**) current-voltage-luminescence curve, (**b**) current efficiency and power efficiency according to the current density, (**c**) EQE and (**d**) EL spectra at 10 mA/cm$^2$; and device photos of 0.2 wt% rubrene (top) and 0.4 wt% rubrene (bottom).

## 4. Conclusions

New sky-blue-light-emitting materials were synthesized by introducing a bulky side group into fused chromophore with anthracene and spirofluorene; these compounds have high LE. The TP-AFF-TP device showed a CIE value of (0.25, 0.45) and achieved an LE of 2.55 cd/A at an applied current density of 10 mA/cm$^2$. The TP-ASF device demonstrated a notable LE of 5.17 cd/A at 10 mA/cm$^2$, along with a CIE value of (0.17, 0.31). The results revealed that TP-ASF exhibited superior LE and a more favorable CIE value compared to TP-AFF-TP. These results show that when a side group is substituted into the fused core, it is better for the high efficiency to use the electron lobe position of core moiety. Furthermore, TP-ASF displays excellent solubility in commonly used organic solvents due to its highly twisted molecular structure. The white OLED device fabricated using the solution process with TP-ASF achieved a LE of 3.13 cd/A and a PE of 2.69 lm/W. Especially, the CIE value and the EL spectra were (0.30, 0.34) and 463, 549 nm, which are matched with the color of lighting application.

**Supplementary Materials:** The following supporting information can be downloaded at: https://www.mdpi.com/article/10.3390/app131810154/s1, Table S1: Time-dependent density functional theory calculations; Figure S1: DSC curve (a) TP-AFF-TP, (b) TP-ASF; Figure S2: Device lifetime of TP-AFF-TP and TP-ASF; Figure S3: $^1$H-NMR Spectrum of 5-Bromoanthraquinone (**1**); Figure S4: $^1$H-NMR Spectrum of 1-Bromoanthracene (**2**); Figure S5: $^1$H-NMR Spectrum of 1-(2-Bromophenyl)anthracene (**4**); Figure S6: $^1$H-NMR Spectrum of 2,7-Dibromo-spiro-fluorene[3,4]naphthalene (**5**); Figure S7: $^1$H-NMR Spectrum of Spiro-fluorene[3,4]naphthalene (**6**); Figure S8: $^1$H-NMR Spectrum of Spiro-fluorene[3,4]-5-bromonapthalene (**7**); Figure S9: $^1$H-NMR Spectrum of 2,7-Bis-[1,1';3',1"]terphenyl-spiro-fluorene[3,4]naphthalene (TP-AFF-TP); Figure S10: $^{13}$C-NMR Spectrum of 2,7-Bis-[1,1';3',1"]terphenyl-spiro-fluorene[3,4]naphthalene (TP-AFF-TP); Figure S11: $^1$H-NMR Spectrum of Spiro-fluorene[3,4]-5-terphenylnaphthalene (TP-ASF); Figure S12: $^{13}$C-NMR Spectrum of Spiro-fluorene[3,4]-5-terphenylnaphthalene (TP-ASF).

**Author Contributions:** Conceptualization, S.K. and J.P.; Methodology, S.P. (Sangwook Park); Validation, S.P. (Sangwook Park), S.K. and S.P. (Sunwoo Park); Formal analysis, S.P. (Sangwook Park), H.L., K.L. (Kiho Lee) and J.P.; Investigation, S.K., S.P. (Sunwoo Park) and H.K. (Hyukmin Kwon); Resources, J.P.; Writing–original draft, S.P. (Sangwook Park); Writing–review & editing, H.L., K.L. and J.P.; Visualization, H.K. and H.L.; Supervision, J.P.; Project administration, J.P.; Funding acquisition, J.P. All authors have read and agreed to the published version of the manuscript.

**Funding:** This research was supported by the Basic Science Research Program through the National Research Foundation of Korea (NRF) funded by the Ministry of Education (2020R1A6A1A03048004). This work was supported by the Technology Innovation Program (20017422, Development of raw materials and process for ArF-i photoresist) funded By the Ministry of Trade, Industry & Energy (MOTIE, Republic of Korea). This work was supported by the Technology Innovation Program (20017832, Development of TiN-based electrode materials and ALD equipment for 10-nm DRAM capacitor electrode deposition process) funded By the Ministry of Trade, Industry & Energy (MOTIE, Korea). This work was partly supported by the GRRC program of Gyeonggi province [(GRRCKYUNGHEE2023-B01), Development of ultra-fine process materials based on the sub-nanometer class for the next-generation semiconductors]. This research was supported by Basic Science Research Capacity Enhancement Project through Korea Basic Science Institute (National research Facilities and Equipment Center) grant funded by the Ministry of Education (No. 2019R1A6C1010052).

**Institutional Review Board Statement:** Not applicable.

**Informed Consent Statement:** Not applicable.

**Data Availability Statement:** Not applicable.

**Conflicts of Interest:** The authors declare no conflict of interest.

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
