# Peer review of "Novel Fused Core Chromophore Incorporating Spirofluorene and Anthracene Groups for Sky-Blue Emission and Solution-Processed White Devices"

_applsci, doi:10.3390/app131810154_

Round 1
Reviewer 1 Report
Sangwook Park and co-workers synthesized two novel molecules based on a fused core comprising anthracene (TP-AFF-TP) and spirofluorene (TP-ASF). These materials exhibited impressive properties, including: 1) emitting a sky-blue light, and 2) demonstrating high thermal resistance exceeding 400°C. Notably, compound TP-ASF exhibited a more distinct sky-blue emission (CIE = 0.17, 0.31), possessed a good luminance efficiency of 5.17 cd/A, and displayed high solubility in common organic solvents. Additionally, the authors successfully fabricated a white OLED device by combining TP-ASF with a yellow emission material, specifically rubrene, through a solution-based process. The manuscript provides an inclusive dataset characterizing the two new materials and the white OLED device performance (CIE = 0.30, 0.34). Before publication, certain corrections must be however addressed:
- In the introduction's background, the authors mentioned the considerable demand for solution-processable OLED devices. However, there is a lack of pertinent references on page 2, lines 46-49. I suggest citing relevant studies to emphasize the impact of this paper to the readers.
- Traditionally, vapor deposition is the mainstream method for OLED device manufacturing due to its guaranteed high performance. Does the solution-based process ensure comparable performance for OLED devices regardless of the materials employed?
- Regarding the molecular design, the incorporation of bulky side groups into the fused core aims to prevent π-π stacking interactions. Presumably, this modification enhances solubility and disrupts the formation of excimers, among other effects. Nevertheless, I recommend providing a brief explanation for clarity.
- Similarly, it would be beneficial for the authors to explain the reason behind introducing a twisted structure to enhance solubility in solution and to potentially cite relevant literature. For example, a binaphthyl structure with a twisted backbone along an auxiliary axis exhibits good solubility, even in the case of polymers bearing a binaphthyl moiety.
- How does the device stability under a fixed current density, considering performance over time (performance vs time)?
- What are the values of photoluminescence quantum efficiency (PLQE) in solution and solid state?
- What are the values of internal quantum efficiency (IQE) in solution and solid state?
- It would be advisable to revise the aspect ratio and resolution in Scheme 1 for improved clarity.
- I recommend referencing the pioneering work on the first fabrication of a multilayer OLED device by Kido et al. (Science, vol. 267, 1333, 1995).
- Including real photos of the sky-blue and white emissions in the current study would make a more profound impression on the readers.
Reviewer 2 Report
Comments to Authors:
Park et al., developed new sky-blue light emitting materials (TP-ASF and TP-AFF-P) by introducing a bulky side group into fused chromophore with anthracene and spiro-fluorene. The results revealed that TP-ASF exhibited superior luminance efficiency and a more favorable CIE value compared to TP-AFF-TP. The reviewer has feeling that this manuscript is well organized and can be accepted by addressing the following minor comments.
1. NMR spectra’s are missing. 13C NMR also should be provided.
2. DSC should be measured and provide the glass transition temperature.
3. The Detailed DFT calculations should be provided for target molecules, instead of core. Analyze the excited state analysis also.
4. Why the ELmax is opposite compared to film state emission values?
5. Authors mentioned in introduction: “Creating blue-light emitters with both high efficiency and extended device lifetimes poses a challenge due to their inherently wide band gap.” Howver there are many reports for blue emitters and current trend also going on by using multiple resonance emitters and Hyperfluorescence blue OLEDs. It would be better to add some information about current trends also. Some refrences can be considered:https://doi.org/10.1016/j.orgel.2022.106501 ; Nat. Photonics 15, 203–207 (2021); https://doi.org/10.1016/j.dyepig.2022.110391; Adv. Funct. Mater. 2022, 32, 2110356.
It is fine.
